# Possible Action of Olaparib for Preventing Invasion of Oral Squamous Cell Carcinoma In Vitro and In Vivo

**DOI:** 10.3390/ijms23052527

**Published:** 2022-02-25

**Authors:** Nanami Nakamura, Hisako Fujihara, Koji Kawaguchi, Hiroyuki Yamada, Ryoko Nakayama, Masaaki Yasukawa, Yuta Kishi, Yoshiki Hamada, Mitsuko Masutani

**Affiliations:** 1Department of Oral and Maxillofacial Surgery, School of Dental Medicine, Tsurumi University, 2-1-3 Tsurumi, Tsurumi-ku, Yokohama 230-8501, Japan; nakamura-n@tsurumi-u.ac.jp (N.N.); kawaguchi-k@tsurumi-u.ac.jp (K.K.); yasukawa-masaaki@tsurumi-u.ac.jp (M.Y.); kishi-y@tsurumi-u.ac.jp (Y.K.); hamada-y@tsurumi-u.ac.jp (Y.H.); 2Department of Oral Hygiene, Tsurumi Junior College, 2-1-3 Tsurumi, Tsurumi-ku, Yokohama 230-8501, Japan; 3Division of Maxillofacial Surgery, Department of Oral and Maxillofacial Surgery, School of Dentistry, Iwate Medical University, 19-1 Uchimaru, Morioka 020-8050, Japan; yamadah@iwate-med.ac.jp; 4Department of Pathology, School of Dental Medicine, Tsurumi University, 2-1-3 Tsurumi, Tsurumi-ku, Yokohama 230-8501, Japan; nakayama-r@tsurumi-u.ac.jp; 5Department of Frontier Life Science, Graduate School of Biochemical Science, Nagasaki University, 1-7-1 Sakamoto, Nagasaki 852-8588, Japan; mmasutan@nagasaki-u.ac.jp; 6Division of Chemotherapy and Clinical Cancer Research, National Cancer Center Research Institute, 5-1-1 Tsukiji, Chuo-ku, Tokyo 104-0045, Japan

**Keywords:** olaparib, oral cancer, epithelial–mesenchymal transition, invasion, metastasis

## Abstract

Despite recent advances in treatment, the prognosis of oral cancer remains poor, and prevention of recurrence and metastasis is critical. Olaparib is a PARP1 inhibitor that blocks polyADP-ribosylation, which is involved in the epithelial–mesenchymal transition (EMT) characteristic of tumor recurrence. We explored the potential of olaparib in inhibiting cancer invasion in oral carcinoma using three oral cancer cell lines, HSC-2, Ca9-22, and SAS. Olaparib treatment markedly reduced their proliferation, migration, invasion, and adhesion. Furthermore, qRT-PCR revealed that olaparib inhibited the mRNA expression of markers associated with tumorigenesis and EMT, notably Ki67, Vimentin, β-catenin, MMP2, MMP9, p53, and integrin α2 and β1, while E-Cadherin was upregulated. In vivo analysis of tumor xenografts generated by injection of HSC-2 cells into the masseter muscles of mice demonstrated significant inhibition of tumorigenesis and bone invasion by olaparib compared with the control. This was associated with reduced expression of proteins involved in osteoclastogenesis, RANK and RANKL. Moreover, SNAIL and PARP1 were downregulated, while E-cadherin was increased, indicating the effect of olaparib on proteins associated with EMT in this model. Taken together, these findings confirm the effects of olaparib on EMT and bone invasion in oral carcinoma and suggest a new therapeutic strategy for this disease.

## 1. Introduction

According to WHO (World Health Organization) reports, the annual number of new oral cancer cases is estimated to be approximately 657,000, with 330,000 deaths [1]. The incidence of oral cancer is affected by tobacco and alcohol consumption [2], as well as human papillomavirus infection [3,4].

Recent standard therapies for oral cancer consist of three major treatments: surgery, radiotherapy, chemotherapy, and combinations of these [5]. Although new therapeutic strategies, such as immunotherapy and targeted therapy, have recently been developed [6,7,8], overall survival has not significantly increased in the past three decades. In advanced head and neck cancer, tumors exhibit invasion of the surrounding tissues, including the mandible and muscles; thus, a wider area is affected by tumor development, and larger resection that requires additional reconstructive surgery is necessary. The morphological structure may be recovered to some extent [9,10,11]; however, oral dysfunctions such as malocclusion, dysphasia, dysarthria, and esthetic dissatisfaction are unavoidable.

Inhibition of poly ADP-ribosylation was recently reported to have anticancer effects [12,13]. PARP1 is an essential enzyme for base excision repair [14], and loss of PARP activity indirectly promotes the accumulation of DNA double-strand breaks [15,16]. The primary mechanism of PARP inhibitors is to competitively block polyADP-ribosylation, which is involved in many cellular processes, including transcriptional regulation and differentiation, as well as DNA repair [17,18], cell death [19,20], telomere regulation [21], and genomic stability [22,23]. In addition to these reports, we recently reported that a clinical PARP1 inhibitor, olaparib, has synergetic effects with cisplatin in vitro and enhances its suppressive effects against the growth of tumor xenografts in vivo [24], suggesting that PARP inhibitors could be favorable for oral cancer therapy in combination with conventional chemotherapy.

To increase the survival rate after initial treatment, prevention of local recurrence and metastasis to other organs are two major factors [25]. Therefore, neoadjuvant chemotherapy or radiotherapy should be considered during follow-up, especially for patients whose margins were closer than expected [26,27]. However, the epithelial–mesenchymal transition (EMT) and the mesenchymal–epithelial transition (MET) are considered to be important for local recurrence. Poly ADP-ribosylation has been reported to be involved in the EMT process, playing a pivotal role depending the circumstances [23,28], and is probably involved in MET, which is the reverse mechanism of EMT. However, to the best of our knowledge, it has not been evaluated in cells derived from oral cancers in vitro or in vivo. In this study, we revealed the role of olaparib in preventing cancer invasion in vitro and its suppressive effects on osteoclasts in vivo. Our results provide an additional role for PARP inhibition in oral cancer therapy to prevent invasion or metastasis, especially during postoperative follow-up.

## 2. Results

### 2.1. Effect of Olaparib on Cell Proliferation

The effect of olaparib on cell proliferation was analyzed. There were no significant differences in the proliferation rate of each cancer cell line, with or without olaparib, until day 5 of culture. However, all three cell lines showed significantly attenuated proliferation with olaparib after day 6 of culture compared with the control group (Figure 1).

### 2.2. Effect of Olaparib on Cell Migration

To investigate the effect of olaparib on cell migration, a wound healing (scratch motility) assay was performed. During the first 4 h of cell migration, the cell migration rate of each cell line did not show a significant difference; however, cell migration was drastically attenuated from 5 h after the initiation of cell migration. The time of scratch closure (more than 90% migration) of each cell line with olaparib was 8.5 h, >12 h, and 10.8 h in Ca9-22, SAS and HSC-2 cell lines, respectively. In the control group, it was 5.5 h, 10.2 h, and 8.4 h in Ca9-22, SAS, and HSC-2, respectively (Figure 2).

### 2.3. Effect of Olaparib on Cell Invasion

The effect of olaparib on cell invasion was examined using three-dimensional cultures. Cells were treated with or without olaparib and then allowed to invade neutralized type I collagen gel for 7 days. Three-dimensional invasion of each cell line was significantly attenuated by olaparib (Figure 3). Moreover, the invasion pattern of each cell line showed distinctive patterns. Ca9-22 showed a relatively collective invasion pattern, while SAS and HSC-2 showed a multicellular invasion pattern, which might indicate that the cell lines were invasive. As shown in Figure 3, olaparib significantly inhibited the invasion of all three cell lines.

### 2.4. Effect of Olaparib on Cell Adhesion

The effect of olaparib on cell adhesion was analyzed using culture plates coated with either collagen I or laminin. As shown in Figure 4, adhesion of all three cell lines to collagen I was significantly increased with 1 µM olaparib treatment (Figure 4a–c). However, adhesion to laminin-coated culture plates showed different patterns. Adhesion of Ca9-22 to laminin was significantly attenuated, while that of SAS and HSC-2 was significantly increased, as observed with collagen-I-coated plates (Figure 4d–f).

### 2.5. Effects of Olaparib on Markers Related to Migration, Invasion, and Adhesion

We examined the effects of olaparib on the mRNA expression of markers related to migration, invasion, and adhesion, Ki67, Vimentin, E-Cadherin, β-catenin, MMP2, MMP9, p53, and integrin α2 and β1, in all three cancer cell lines by quantitative real-time PCR (Figure 5). The expression levels of Ki67, Vimentin, β-catenin, MMP2, MMP9, p53, and integrin α2 and β1 in all three cell lines were significantly suppressed in the olaparib group compared with those in the control group after 5 days’ exposure to 1 µM olaparib. However, the expression level of E-Cadherin was significantly augmented with olaparib treatment in Ca9-22 and slightly increased in HSC-2. These results suggest that olaparib treatment could suppress the expression levels of mRNA markers related to migration, invasion, and adhesion. The increase in E-Cadherin in HSC-2 and Ca9-22 may be associated with both EMT suppression and MET promotion.

### 2.6. Radiographic Analysis of the In Vivo Effects of Olaparib on Xenograft Tumor Growth and Bone Destruction by Tumor Invasion

To analyze the effects of olaparib in murine oral-grafted tumor models, tumor xenografts were generated by injection of HSC-2 cells (1.3 × 10^5^ cells) into masseter muscles. Ca9-22 and SAS cell lines were also grafted, resulting in growth stop at 5 mm diameter. Tumor volumes of the control group increased during the experimental period, whereas the tumor growth of the olaparib groups was significantly suppressed compared with the control group from 19 days after tumor cell injection. The average estimated tumor size in the control group was significantly larger (1.6 times) than the olaparib-treated group, although the estimated tumor size varied (Figure 6a).

On day 63 of tumor cell injection, all mice were analyzed by micro-CT before they were sacrificed under general anesthesia. When mice with similar lateral tumor growth were compared with or without olaparib treatment, the bone condition inside the tumors was drastically different between the two groups. In the control group, the zygoma arch, ramus, mandibular angle, and external auricular canal were severely destroyed by tumor invasion (Figure 6b–d). However, bone destruction was drastically suppressed by olaparib and only a small area of external cortical bone of the mandibular ramus was destroyed (Figure 6e).

### 2.7. Histopathological Analysis of the In Vivo Effects of Olaparib on Xenograft Tumor Growth and Bone Destruction by Tumor Invasion

We next analyzed xenograft tumor growth and bone destruction microscopically. HSC-2 tumors of the control group showed more severe invasion to the mandible compared with that of the olaparib-treated group (Figure 7a). To quantify the intensity of DAB staining for Ki67 staining in xenograft tumor area, photos of all categories were analyzed by ImageJ Fiji. There was a tendency that the Ki67 intensity of category 0 was less than categories 1–3; however, significant difference was not observed within each category. To analyze tumor invasion levels towards the mandible, we defined the bone invasion level of tumors as one of four categories: 0—non-invasion to the buccal cortical bone of the mandible; 1—invasion to the buccal cortical bone of the mandible; 2—invasion to the lingual cortical bone of the mandible; and 3—invasion to the suprahyoid muscles. The numbers of tumors of categories 0, 1, 2, and 3 were 0, 0, 8, and 6 in the control group, and 0, 9, 2, and 3 in the olaparib-treated group, respectively (Figure 7c). Therefore, olaparib significantly suppressed tumor invasion to the mandible in vivo (*p* < 0.01) (Figure 7d).

### 2.8. Olaparib Affected the Expression of Proteins Involved in Invasion and Adhesion in Tumor Tissues

To assess the mechanism of the effects of olaparib on tumor invasion to the mandible, we analyzed the protein levels of factors related to osteoclast activation, RANK and RANKL, and factors related to EMT, E-cadherin, N-cadherin, Twist, SNAIL, and PARP1, using HSC-2-derived tumors resected on day 60. As shown in Figure 8, the protein levels of RANK, RANKL, SNAIL, and PARP1 in the olaparib group were significantly reduced compared with those of the control group (*p* < 0.01). However, that of E-cadherin was significantly increased in the olaparib group compared with that of the control group (*p* < 0.05). The protein levels of N-cadherin and Twist in both groups did not show any significant differences.

## 3. Discussion

In this study, olaparib showed a significant reduction on the in vitro invasion of cancer cell lines derived from oral carcinoma, HSC-2, Ca9-22, and SAS, and a significant reduction in in vivo xenograft HSC-2 tumors, especially for bone invasion. We previously reported the synergetic effects of olaparib with cisplatin on oral carcinoma-derived cell lines in vitro and in vivo [24]. In this report, the same cell lines were used in the in vitro study, and we also attempted to create tumor xenografts using these three cell lines. However, only HSC-2 could consistently develop tumor xenografts. Moreover, we attempted to establish a lung metastasis model of oral carcinoma-derived cell lines but were not successful. Currently, many lung metastasis models use melanoma-derived cell lines [29,30]. We investigated the factors related to EMT/MET and osteoclastogenesis in which PARP inhibitor attenuated or stimulated the expression levels in vitro and in vivo.

EMT plays a role in cancer development and progression by converting static epithelial cells into migratory and microenvironment-interacting mesenchymal cells, and by modulating the chemoresistance and stemness of tumor cells. Several findings highlighted that both EMT and MET contribute to cancer invasion and metastasis [31]. During EMT and MET, the expression profile of epithelial and mesenchymal markers E-cadherin, N-cadherin, Vimentin, cytokeratins, tight junction proteins, and β-catenin, are partly revealed [31,32,33,34,35]. The role of PARP1 in EMT and MET remains to be fully determined, but has been reported to have a pivotal role in EMT, as well as other roles in DNA repair and genomic stability [23,32,36].

We predicted that olaparib treatment would prevent invasion and metastasis according to our previous report [24], but it would not have been unexpected if the role of PARP1 went either way. The most unexpected data were the results of the cell adhesion assay, which showed significantly increased adhesion to three cell lines with 1 µM olaparib treatment (Figure 4a–c,e,f) except the decreased adhesion of Ca9-22 to laminin (Figure 4d). This could be attributed to the differences in the extracellular matrix regarding collagen I and laminin [37,38], and the affinity between the extracellular matrix and each cell line. Moreover, considering that the process of MET is required for cells migrating to secondary sites to colonize metastatic regions [39], and genes increased in MET are reported to be E-cadherin and tight junction proteins [40], the results of the cell adhesion assay were supported by the significantly increased level of E-cadherin in Ca9-22 and HSC-2 cells (Figure 5). The second most unexpected results were the result of cell proliferation assay, which showed suppressive effect six days after 1 µM olaparib exposure and SAS and HSC-2 cell lines did not reach confluence (Figure 1b,c). The mechanism of suppression of cellular proliferation under Olaparib exposure condition could be hypothesized as follows; First, Parp inactivation by Olaparib cause DNA double-strand break, which is proved by γ-H2AX expression [41]. Gamma-H2AX is also one of the autophagy markers that regulate cell cycle in cancer cells [41]. Then, accumulation of DNA damage will cause delayed suppressive effects on cellular proliferation; although, cell cycle arrest (especially G2/M arrest) was recovered within 48 h after removing olaparib, and the cytotoxicity of 1 µM olaparib was not statistically significant [24].

Nonetheless, the expression pattern of the significantly decreased expression of E-cadherin would be the trigger to start EMT, which is downregulated by many factors, such as SNAIL, ZEB, and Twist [32]. The decreased cell migration activity (Figure 2a–f) and invasion activity (Figure 3a–f) in all three cell lines in vitro was considered to the result of EMT suppression. Moreover, the mRNA expression level of tumorigenesis-related factors Ki67, β-catenin, and p53, and EMT-related factors Vimentin, MMP2, MMP9, and *integrin α2* and *β1* were all attenuated by 1 µM olaparib treatment in all three cell lines, which was consistent with previous studies that showed upregulation of E-cadherin [36] and downregulation of MMP-2 and MMP-9 [42,43], Vimentin [44], and integrin α2 and β1 [45,46,47]. From the in vitro results, olaparib could contribute to suppression of tumor invasion, and the different olaparib effects on E-cadherin in the SAS cell line would be the possible clue to the pivotal role of PARP1 activity in EMT.

In vivo analysis using tumor xenografts showed significantly reduced tumorigenesis in the olaparib-treated group. To exclude the effect of olaparib on tumor growth, we compared similar volume tumors (Figure 6b–e) and found aggressive bone invasion in the control group. Moreover, tumors’ inward invasion levels were significantly reduced in the olaparib-treated group (Figure 7a–c). The data indicate that the mechanism of the significant reduction in tumor invasion in the olaparib-treated group could exclude cytotoxicity in our experimental conditions. The expression levels of factors related to EMT/MET were significantly regulated according to previous reports showing upregulation of E-cadherin [36] and Twist [32], and downregulation of SNAIL [36,48]. In our study, the expression of N-cadherin did not show any significant differences following olaparib treatment, which was reported to be both upregulated [49] and downregulated by PARP1 inhibitor [44].

Considering both in vitro and in vivo results, we should notice another concept of anaplastic transition (APT). In the invasion front of carcinoma, there are CK and Vimentin highly expressed epithelial cancer cells, which have contact with cancer-associated fibroblasts (CAFs) and affect each other. Then, these epithelial cancer cells will dedifferentiate into a more primitive status called APT [50]. APT is suggested to have an association with local recurrence of oral tongue squamous cell carcinoma; therefore, olaparib may have the potency to directly control local recurrence and invasion because of Vimentin attenuation by olaparib (Figure 5).

Furthermore, the expression level of osteoclastogenesis proteins RANK and RANKL were also downregulated by olaparib, which resulted in the prevention of bone invasion. This is inconsistent with a recent study that showed that olaparib increased breast cancer bone metastasis via PARP2 but not PARP1 using myeloid cells. Olaparib inhibition or *PARP1/PARP2* deletion promotes osteoclast differentiation and bone loss with impairment of CCL3 expression through enhancing the transcriptional repression of β-catenin [51]. Moreover, a pivotal role of PARP1 on osteoclastogenesis was also reported in the basal repression of genes that are upregulated during RANKL-induced osteoclastogenesis using 3-aminobenzamide [52].

Regarding osteoblastogenesis, a critical role of PARP1 was also previously reported. Our previous study demonstrated that PARP inhibitor PJ34 attenuated mesenchymal osteoblast differentiation via the BMP-2 signaling pathway in vitro [53], which suggested a progressive effect on osteoclastogenesis; furthermore, another study showed that tankyrase inhibitor XAV-939 enhances osteoblastogenesis and mineralization of human skeletal mesenchymal stem cells via TNF, NF-κB, and STAT signaling [54]. Thus, the effect of olaparib is suggested to be pivotal on other pathways, or this role could be due to the differences in the derived cell lines.

Although much in this field remains to be investigated, this initial report directly showed the effects of the clinical use of olaparib on the EMT/MET pathway and on bone invasion in cell lines derived from oral carcinoma and in an orthotopic xenograft model. In this experiment, we only used Olaparib to evaluate the effects of Parp activity deterioration on EMT/MET pathway and osteogenic invasion; however, the combinatorial effects of olaparib and antitumor agents such as cisplatin would be required to proceed to the next step for clinical usage. Considering the crucial role of olaparib, our findings could provide a new therapeutic strategy for oral carcinoma—especially for neoadjuvant chemotherapy to prevent microinvasion, local recurrence, and metastasis to other organs via regulation of the EMT/MET pathway—and aid the examination of current PARP inhibitors for bone metastasis and bone loss.

## 4. Materials and Methods

The ethical committee of our institution approved all animal experimental procedures according to the guidelines on animal experiments of Tsurumi University School of Dental Medicine on 3 July 2012 (no. 12040).

### 4.1. Cell Culture of Cell Lines Derived from Oral Carcinoma

Three cell lines were used to evaluate the various origins of oral SCC: HSC-2 from oral squamous cell carcinoma in a 69-year-old man, Ca9-22 from gingiva squamous cell carcinoma in a man, and SAS from tongue squamous cell carcinoma. All cell lines were obtained from RIKEN, Tsukuba, Japan, in 2012. Growth medium consisted of 10% fetal bovine serum (Biowest, Nuaille, France), 100 U/mL penicillin, and 100 µg/mL streptomycin (Sigma-Aldrich, St. Louis, MO, USA) in minimum essential Eagle’s medium (Sigma-Aldrich, St. Louis, MO, USA) for HSC-2 and Ca9-22 cells, and in Dulbecco’s modified Eagles medium (Sigma-Aldrich, St. Louis, MO, USA) for SAS cells. The cell culture conditions were as follows: 37 °C with 5% CO_2_, growth medium exchange every 3 days, and passaged at 1:5 after reaching confluency.

### 4.2. Proliferation Assay

All cell lines were seeded into 6-well plates at a density of 2 × 10^3^ cells/well in triplicate with growth medium containing 0 or 1 µM olaparib. The used olaparib concentration was decided according to previous studies including the result of MTT assay of our previous study [24,55]. The culture medium was changed every 3 days. Cell numbers were counted with a hemocytometer.

### 4.3. Wound Healing Assay

The wound healing assay (scratch assay) was performed as previously described [56]. First, 2 × 10^4^ cells were seeded into 6-well plates and cultured in growth medium until confluence. Then, floating cells were removed by washing twice with PBS. Cells were cultured with medium containing 12 µg/mL mitomycin C (Kyowa Hakko Kirin Co., Tokyo, Japan) for 2 h 15 min at 37 °C with 5% CO_2_ to inhibit cell proliferation. After mitomycin C treatment, a vertical wound was created using a spatula across the cell monolayer. The medium and debris were aspirated and replaced with fresh culture medium with or without 1 µM olaparib. Next, cell migration was visualized with a microscope (BX51 System Microscope (Olympus, Tokyo, Japan)) every 2 h. An image of the same region was captured by a digital microscope camera (DP70; Olympus, Tokyo, Japan) at every time point, and the areas without cells were measured by Image J software (NCI, Bethesda, MD, USA).

### 4.4. Invasion Assay

Invasion assay was performed as described previously [57,58]. First, TIG-1-20 (JCRB0501) human embryonic pulmonary fibroblast cells were mixed into neutralized type I collagen gel (Cellmatrix type I-A, (Nitta Gelatin Inc., Osaka, Japan)) according to the manufacturer’s instructions. The solution was added to 6-well plates (3 mL/well) and allowed to harden for 30 min in a 5% CO_2_ incubator at 37 °C in humidified air. Then, the 3 cell lines were dispensed onto each gel at a density of 1 × 10^6^ cells in 3 mL 3D culture medium with or without olaparib followed by overnight incubation. The next day, the hardened gels were detached from the plates and incubation was continued for 1 week until the size of the contracted gel stabilized. Cell strainers (Beckton, Dickinson and Company, NJ, USA) were placed upside-down into a fresh 6-well plate after removing the strainer handles using sterilized scissors, and the 3D culture medium was poured into the wells until the nylon mesh of the strainers was covered. The contracted gel discs were then placed onto the mesh so that the oral cancer cells laid on the top of gel discs, and the fluid level was adjusted to just below the upper edge of the gel. During the experiment, half of the culture fluid was renewed every other day. After 1 week of air–liquid interface culture, the gel discs were fixed in phosphate-buffered 10% formalin solution overnight and embedded in paraffin, and then vertical sections were stained with hematoxylin and eosin.

### 4.5. Adhesion Assay

The 96-well culture plates (Falcon, Corning, NY, USA) were coated with 50 µL type I collagen (10 µg/mL in PBS; Cellmatrix type I-P; Kurabo Industries LTD, Osaka, Japan) and laminin (10 µg/mL; Mouse EHS Tumor, Wako, Osaka, Japan) and incubated for 24 h at 4 °C, and residual liquid in the wells was discarded [47,48]. Then, nonspecific binding sites were blocked with 3% BSA (200 µL/well) for 2 h at room temperature. Next, 5 × 10^6^ cells were plated in triplicate onto wells with both coatings, cultured in serum-free medium with or without olaparib, and allowed to adhere for 4.5 h at 37 °C. Then, nonadherent cells were removed by two gentle washes with PBS. The remaining cells were fixed with 4% paraformaldehyde for 15 min and stained with hematoxylin for 5 min. After washing with tap water, plates were dipped into distilled water at 40 °C for 5 min. Subsequently, plates were washed with distilled water, and glycerinated gelatin was added to plates for 1 h to harden the gel. For the analysis of plate-adhered cells, images of each well were viewed under a BX51 System Microscope (Olympus, Tokyo, Japan). Images were recorded using a digital microscope camera (DP70; Olympus, Tokyo, Japan). All images were converted from color to grayscale, and the luminance level was measured by ImageJ software (NCI, Bethesda, MD, USA) [59].

### 4.6. Total RNA Extraction and Real-Time PCR

Total RNA was isolated from cells using TRIzol reagent (Invitrogen, Waltham, MA, USA) and isolated RNA (5 µg) was reverse transcribed to cDNA using a PrimeScript 1st Strand cDNA Synthesis Kit (Takara, Ohtsu, Shiga, Japan) according to the manufacturers’ protocol. For real-time PCR analyses, a combination of SYBR Premix Ex Taq II (Takara, Ohtsu, Shiga, Japan) and the StepOnePlus Real-Time PCR System (Applied Biosystems, Waltham, MA, USA) was used at the recommended thermal cycling settings: one initial cycle at 95 °C for 30 s, followed by 40 cycles at 95 °C for 3 s and 60 °C for 30 s. Primer details are summarized in Table 1. The quantitative gene expression level of each marker was normalized to *β-actin* expression levels.

### 4.7. Animal Models

The 5-week-old nude mice (Balb/c, nu/nu, male; SLC Co., Shizuoka, Japan), weighing approximately 20 g at 5 weeks old, were used in this study. The mice were maintained under specific pathogen-free conditions throughout the experiments at a constant room temperature with a 12 h night/day cycle, and the mice had ad libitum normal food chow (CE-2; SLC Co., Shizuoka, Japan) and sterilized water. A combination of medetomidine hydrochloride (Nippon Zenyaku Kogyo Co., Fukushima, Japan), midazolam (Astellas Pharma Inc., Tokyo, Japan), and butorphanol (Meiji Seika Pharma Co., Tokyo, Japan) were used for general anesthesia during cell injections [60,61]. HSC-2, Ca9-22, and SAS cells were injected into both masseter muscles (1.3 × 10^5^ cells in 120 µL growth medium per mouse). Among the three cancer cell lines, only HSC-2 cells could stably develop tumors. Therefore, in vivo experiments were performed using HSC-2-cell-derived tumor xenografts.

### 4.8. Experimental Animal Protocol

One week after tumor cell injection, mice were randomly divided into control (200 µL saline) and olaparib (25 mg/kg per body weight, dissolved in 100 µL sterilized water) groups. Then, they were intraperitoneally injected every 3 days until they were sacrificed. Although the administration of olaparib in the clinic is oral, intraperitoneal injection was recommended by the manufacturer because it is easier to manipulate, while oral gavage administration was avoided because of ethical constraints regarding animal welfare. Body weight and tumor size were measured at the time of administration, and the tumor volume was calculated using the following equation [62].
Tumor volume = verticality × width × height × 0.5236

Three days after the last administration, all surviving mice were radiologically analyzed under anesthesia and sacrificed.

### 4.9. Radiological Analysis by Micro-CT

To obtain high-resolution radiographs of tumor invasion to bone, each maxillofacial area was analyzed under general anesthesia using the same method described above [60,61]. Each sample was scanned by micro-CT (InspeXio SMX-100CT; Shimadzu Corporation, Kyoto, Japan), and the CT settings were as follows: tube voltage, 160 kV; tube currency, 70 µA; exposure time, 124.928 ms; and thickness, 0.34 mm. All data were converted to 3D structures with analysis software (TRI/3D-BON; Ratoc System Engineering Co., Tokyo, Japan).

### 4.10. Histopathological Analysis

Mice were treated as described above and then sacrificed on day 63, 3 days after the last administration. The heads of mice with tumors were resected and immediately fixed in 10% formalin/PBS (Wako Pure Chemical Industries, Osaka, Japan). Samples were then decalcified in OSTEOSOFT (Merck KGaA, Darmstadt, Germany) for 1 month, which was replaced every 3 days. Then, samples were immersed in 70% alcohol for 48 h and embedded in paraffin. Tissue sections (4 µm) were mounted on silane-coated slides (New Silane III; Muto Pure Chemicals Co., Tokyo, Japan), deparaffinized with xylene (Wako Pure Chemical Industries, Osaka, Japan), and rehydrated with graded alcohol solutions (Wako Pure Chemical Industries, Osaka, Japan). Finally, the histopathological analysis of the specimens was performed with hematoxylin and eosin staining (Merck KGaA, Darmstadt, Germany).

### 4.11. Immunohistochemistry

After deparaffinization with xylene and rehydration with descending ethanol concentrations, Immunosaver (Nisshin EM, Tokyo, Japan) was used to for antigen retrieval according to the manufacturer’s protocol. In short, sections were incubated in Immunosaver (1:200 dilution in tap water) for 40 min at 95 °C and then transferred to tap water and incubated for 10 min at room temperature. Subsequently, the slides were treated with 3% hydrogen peroxide (Wako Pure Chemical Industries, Osaka, Japan) in methanol (Wako Pure Chemical Industries, Osaka, Japan) to inactivate endogenous peroxidase for 30 min at room temperature. After treatment with 20% normal goat serum (Nichirei Corporation, Tokyo, Japan) for 30 min at room temperature, sections were incubated with the primary antibody at 4 °C overnight. The antibody used in this study was rabbit polyclonal anti-Ki67 antibody (1:50 dilution; Gene Tex Inc., Irvine, CA, USA). Antibody was diluted with PBS (pH 7.4) containing 1% bovine serum albumin (Sigma-Aldrich, St. Louis, MO, USA) and incubated at 4 °C. After several PBS washes, Histofine Simple Stain MAX-PO(MULTI) (Nichirei Corporation, Tokyo, Japan) and DAB (Vector Laboratories, Burlingame, CA, USA) was used to visualize the bound antibody according to the manufacturer’s protocol. The sections were counterstained with hematoxylin and mounted. As a negative control, sections were processed without exposure to the primary antibody. The sections were viewed under a BX51 System Microscope (Olympus, Tokyo, Japan). Images were recorded using a digital microscope camera (DP70; Olympus, Tokyo, Japan).

### 4.12. Quantification of the Intensity of Immunohistochemical Positivity

The photographs of the immunohistochemical analysis with Ki67 were analyzed by ImageJ Fiji (National Institutes of Health, Bethesda, MD, USA, Fiji Downloads. available online: https://imagej.net/Fiji/Downloads (accessed on 12 November 2021)) and the mean intensity of the selected area was calculated. First, all photographs were deconvoluted, and DAB color version images were chosen from divided into three images. Then, the subjected area was selected and cropped. Subsequently, the mean density of DAB color was calculated and converted from max intensity 255 [59].

### 4.13. Western Blot Analysis

Slice of approximately 1 mm at the maximum cutting surface of HSC-2 tumor xenografts of both groups were lysed with RIPA buffer (10 mM Tris-HCl, 1% NP-40, 0.1% SDS, 150 mM NaCl, and 1 mM EDTA) containing protease inhibitor cocktail (Thermo Fisher Scientific, Waltham, MA, USA). Samples was minced and centrifuged at 14,000× *g* for 20 min at 4 °C. Then, NuPAGE LDS sample buffer (Invitrogen, Carlsbad, CA, USA) was added to the supernatant and heated at 98 °C for 5 min. Next, 80 µg protein samples were separated electrophoretically on 8% or 12% Bis-Tris gels by the NuPAGE System and consequently electroblotted onto polyvinylidene difluoride membranes using an iBlot Dry Blotting System (all from Invitrogen, Waltham, MA, USA). The membranes were blocked with 5% skim milk (Wako Pure Chemical Industries, Osaka, Japan) at room temperature for 30 min before immunochemical reaction with primary antibodies dissolved in 2.5% skim milk at 4 °C overnight. The primary antibodies used in this study were anti-RANK (1:250 dilution; Bioss Antibodies, Woburn, MA, USA), anti-RANKL (1:500 dilution; Bioss Antibodies, Woburn, MA, USA), anti-E-cadherin (1:5000 dilution; Gene Tex, Irvine, CA, USA), anti-N-cadherin (1:500 dilution; Bioss Antibodies, Woburn, MA, USA), anti-Twist (1:5000 dilution; Gene Tex, Irvine, CA, USA), anti-SNAIL (1:1500 dilution; Gene Tex, Irvine, CA, USA), anti-PARP1 (1:200 dilution; Santa Cruz Biotechnology, Santa Cruz, CA, USA), and anti-β-actin (1:1000 dilution; Abcam, Cambridge, UK). Subsequently, membranes were incubated with peroxidase-conjugated secondary antibodies at room temperature for 30 min. Then, specific bands were detected by chemiluminescence assay (Amersham RCL Prime Western Blotting Detection Reagent, GE Healthcare, Buckinghamshire, UK). Finally, images were scanned with a C-Digit Scanner and analyzed by Image Studio Lite Software (both from Li-COR Biosciences, Lincoln, NE, USA).

### 4.14. Statistical Analysis

All data were statistically analyzed with independent t-test and Mann–Whitney U test by SPSS v.20.0 (IBM Corp., Armonk, NY, USA).

## Figures and Tables

**Figure 1 ijms-23-02527-f001:**
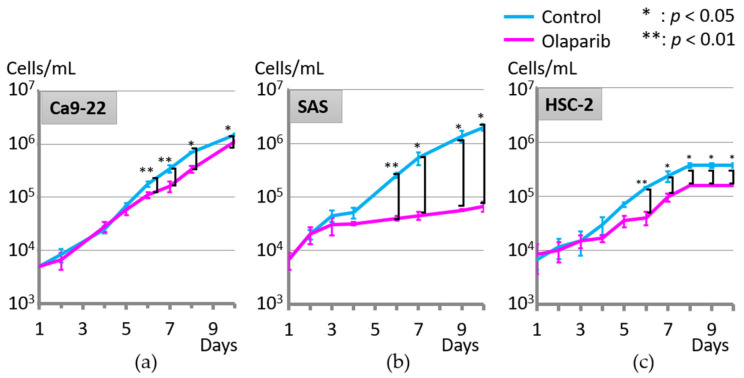
Olaparib at 1 µM significantly attenuated cell proliferation after 6 days of cell culture compared with the control group. (**a**) Ca9-22, (**b**) SAS, and (**c**) HSC-2. *; *p* < 0.05, **; *p* < 0.01. Error bars represent standard deviation.

**Figure 2 ijms-23-02527-f002:**
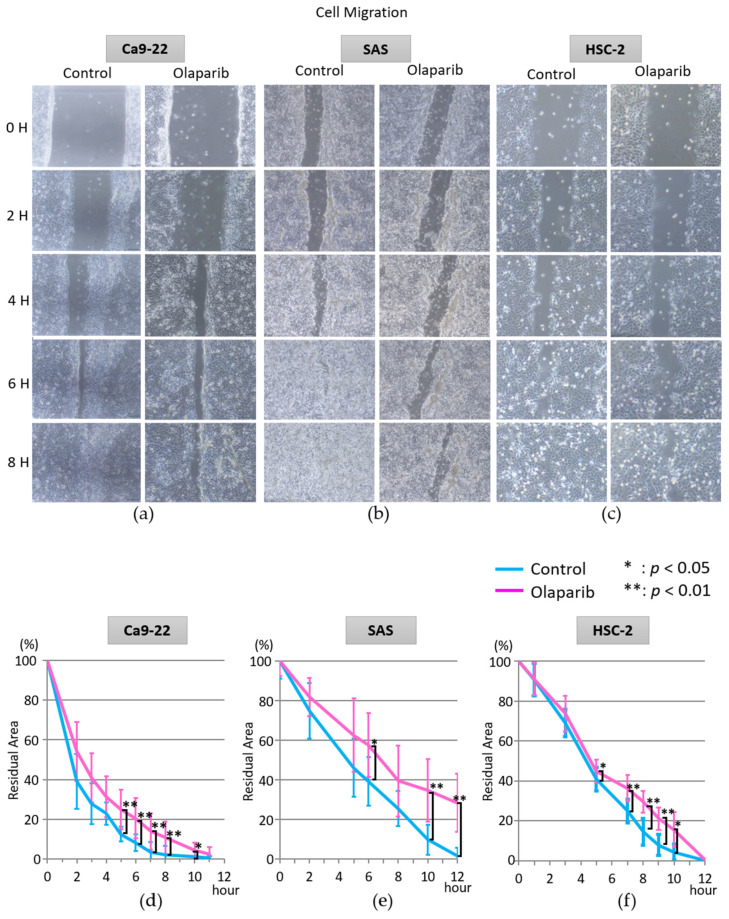
Analysis of wound healing assay in the three cell lines. Representative images of Ca9-22 (**a**,**d**), SAS (**b**,**e**), and HSC-2 (**c**,**f**). Olaparib at 1 µM significantly delayed the migration of these oral SCC cell lines compared with cells without olaparib. Once cell culture reached confluency, a scratch was made using a spatula, and migration was allowed. In the control group, the closure of the scratch (more than 90%) took approximately 6 h, 10 h, and 9h in Ca9-22, SAS, and HSC-2 cell lines, respectively. However, closure of cell scratches (more than 90%) treated with 1 µM olaparib took approximately 9 h, >12 h, and 10.5 h in Ca9-22, SAS, and HSC-2 cell lines, respectively. *; *p* < 0.05, **; *p* < 0.01, Error bars represent standard deviation.

**Figure 3 ijms-23-02527-f003:**
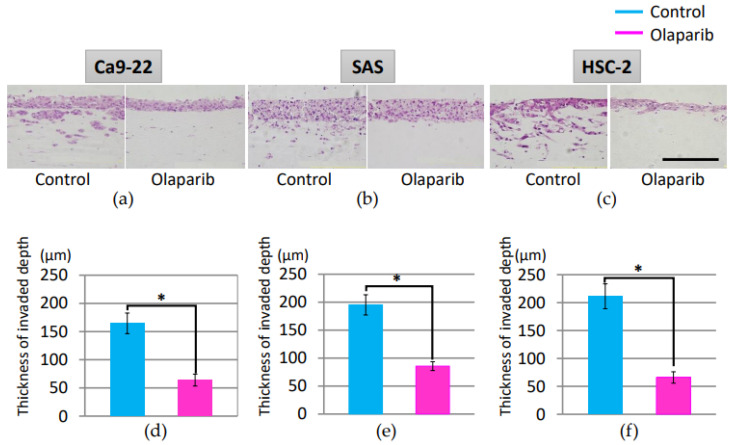
Analysis of organotypic invasion assay of three cancer cell lines. Representative images of Ca9-22 (**a**,**d**), SAS (**b**,**e**), and HSC-2 (**c**,**f**). Olaparib at 1 µM significantly restricted invasion of these oral SCC cell lines compared with cells without olaparib. *; *p* < 0.05. Bar = 200 µm. Error bars represent standard deviation.

**Figure 4 ijms-23-02527-f004:**
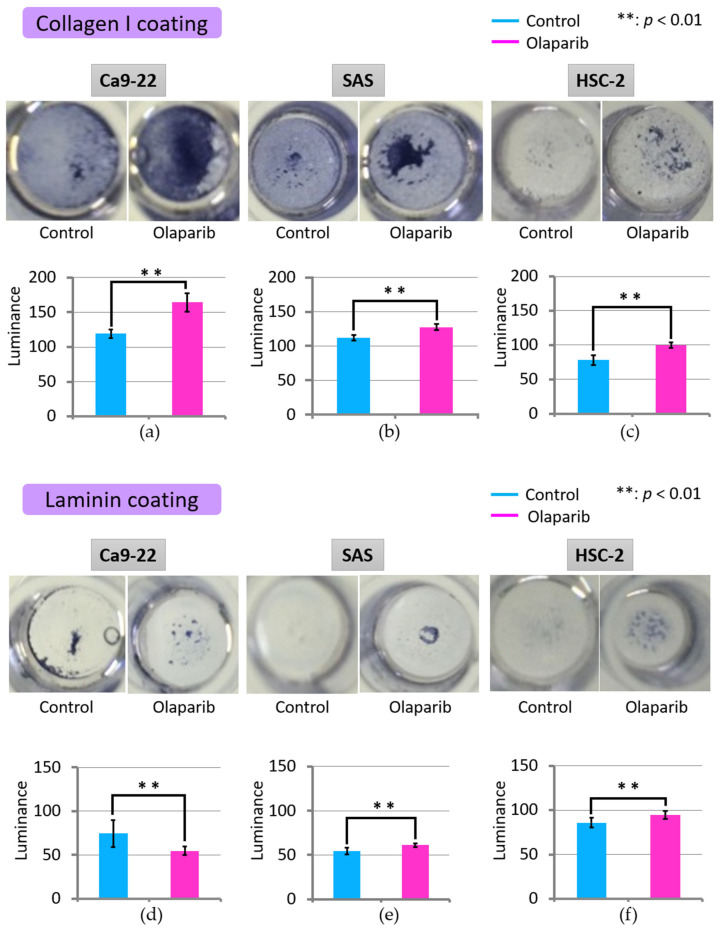
Representative results of adhesion of three cell lines to collagen-I-coated (**a**–**c**) and laminin-coated plates (**d**–**f**). Adhesion of three cell lines to both collagen-coated and laminin-coated plates was significantly increased with olaparib treatment, except Ca9-22 to laminin-coated plates (*p* < 0.01). **: *p* < 0.01. Error bars represent standard deviation.

**Figure 5 ijms-23-02527-f005:**
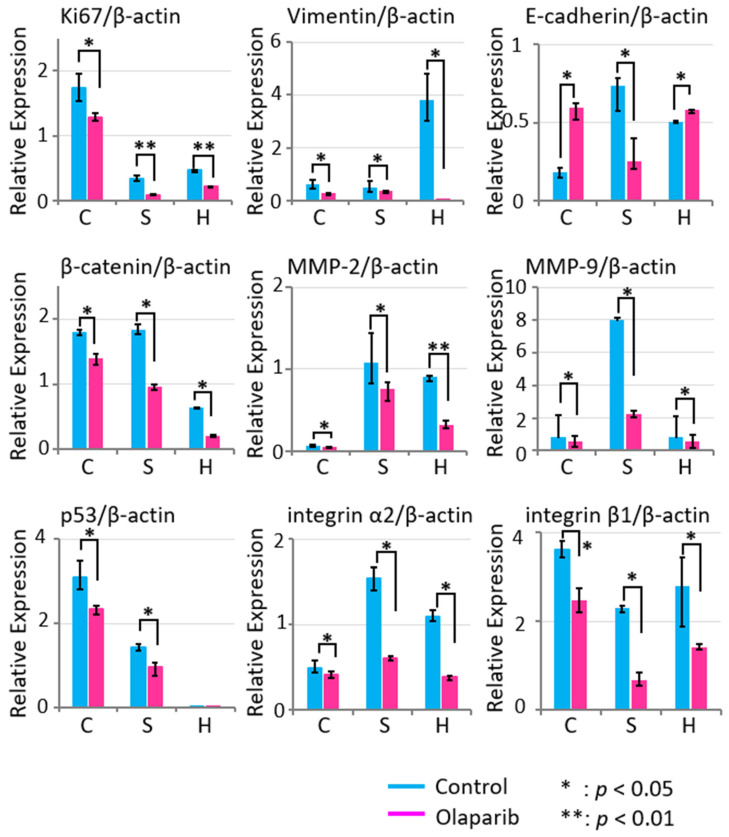
Expression level of mRNA related to migration, invasion, and adhesion in Ca9-22 (C), SAS (S), and HSC-2 (H) cell lines. Expression levels of mRNA related to migration, invasion, and adhesion were suppressed by 1 µM olaparib treatment except E-Cadherin in Ca9-22 and HSC-2 cell lines. *: *p* < 0.05, **: *p* < 0.01. Error bars represent standard deviation.

**Figure 6 ijms-23-02527-f006:**
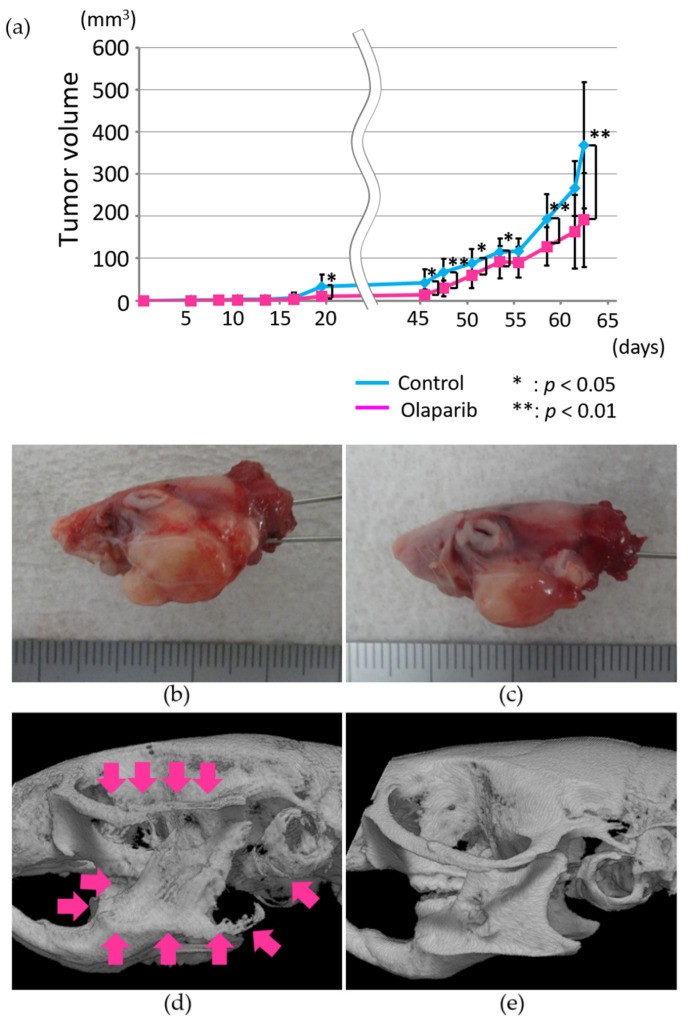
(**a**) Estimated lateral growth of tumors in the control and olaparib groups. Representative image of similar estimated tumor volumes in the control group (**b**) and olaparib group (**c**). From micro-CT images, destruction of the mandible and zygoma by tumor invasion was much more severe in the control group (**d**) than in the olaparib-treated group (**e**).

**Figure 7 ijms-23-02527-f007:**
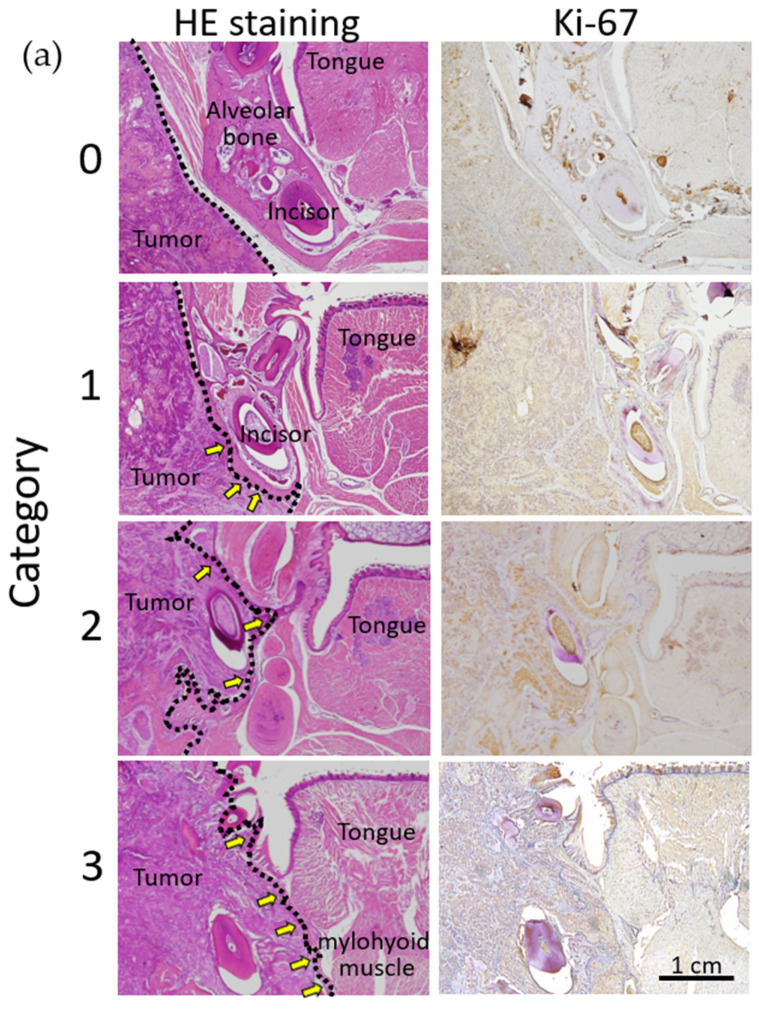
(**a**) Representative image of xenograft tumor invasion to the mandible for each category. Category 0—non-invasion to the buccal cortical bone of the mandible; category 1—tumor invasion to the buccal cortical bone of the mandible; category 2—tumor invasion to the lingual cortical bone of the mandible; and category 3—invasion to the suprahyoid muscles. Black dotted line indicates the frontier line of tumor invasion. As shown in the right column, Ki-67 positivity was stronger in categories 2 and 3 (control group) compared with categories 0 and 1 (olaparib-treated group). Immunostaining of Ki-67 showed evidence of tumor sites (right panels). Bar = 1 cm. (**b**) Semi-quantitative analysis of mean intensity of Ki67 positivity in xenograft tumor areas, and significant difference was not observed within each category. Error bars represent standard deviation. (**c**) Definition of tumor invasion level (upper panel), and the number of tumors of each category in the control group and olaparib group (lower panel). (**d**) Comparison of tumor invasion between the control group and olaparib-treated group. Tumors in the control group showed significantly aggressive invasion to the mandible (*p* < 0.01).

**Figure 8 ijms-23-02527-f008:**
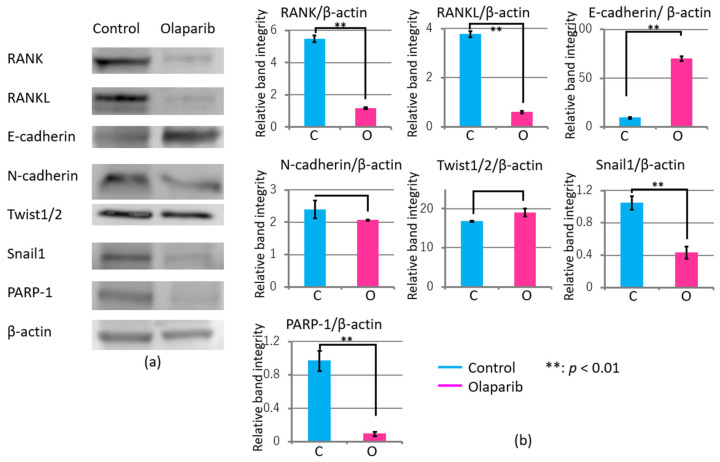
(**a**) Effect of olaparib on osteoclast activity and EMT of tumor xenografts during growth and invasion to the mandible. (**b**) Relative band integrity was normalized by the expression level of β-actin. Values (*n* = 3) are expressed as mean ± SEM. ** *p* < 0.01.

**Table 1 ijms-23-02527-t001:** Sequence of primers used in real-time RT-PCR.

Gene	Length	Forward Sequence (5’ > 3’)	Reverse Sequence (5’ > 3’)
*Ki67*	129	ATCGTCCCAGGTGGAAGAGTT	ATAGTAACCAGGCGTCTCGTGG
*vimentin*	72	AGCCGAAAACACCCTGCAAT	CGTTCAAGGTCAAGACGTGC
*E-cadherin*	196	GGTGCTCTTCCAGGAACCTC	GGAAACTCTCTCGGTCCAGC
*β-catenin*	116	GAGTGCTGAAGGTGCTATCTGTCTG	GTTCTGAACAAGACGTTGACTTGGA
*MMP2*	266	GTGTTCTTTGCAGGGAATGAAT	ACGACGGCATCCAGGTTATC
*MMP9*	80	CGCCCATTTCGACGATGAC	CGCCATCTGCGTTTCCAA
*p53*	200	CTACAAGCAGTCACAGCAC	AGTCAGAGCCAACCTCAG
*Integrin β1*	185	CAAGCAGGGCCAAATTGTGG	CCTTTGCTACGGTTGGTTACATT
*Integrin α2*	224	CTGGAGTGGCTTTCCTGAG	ACTG ATTCCCACATTGCTG
*β-actin*	262	GGCTGTATTCCCCTCCATCG	GTTGGCCTTAGGGTTCAGGG

## Data Availability

This manuscript was reported as the sequel of a previously published manuscript from our institute, as cited in reference [24]: Int J Mol Sci, 2016 24; 17(3):372. doi: 10.3390/ijms17030272. https://www.ncbi.nlm.nih.gov/pmc/articles/PMC4813136/ (accessed on 5 May 2021).

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
