# Peer review of "Possible Action of Olaparib for Preventing Invasion of Oral Squamous Cell Carcinoma In Vitro and In Vivo"

_ijms, 2022, doi:10.3390/ijms23052527_

Round 1
Reviewer 1 Report
The authors report the activity of olaparib on 3 cell lines of oral carcinoma. All the reported reports are adequate, but even though the authors previously reported on the combination of cisplatin and olaparib on the same cell lines, comparisions of olaparib alone vs cis-olaparib are not presented , expecially in the xenograft models. This is not a must, but should be further documented in the discussion. Morevoer, olaparib is reported to induce autophagy. The studies done by the authors are not related to this, but this should be mentionned in the discussion as well.
Otherwise, this paper is adequately written, is not the most original, but adds to the body of knowledge on PARP inhibition in HNSCC.
Author Response
The authors report the activity of olaparib on 3 cell lines of oral carcinoma. All the reported reports are adequate, but even though the authors previously reported on the combination of cisplatin and olaparib on the same cell lines, comparisions of olaparib alone vs cis-olaparib are not presented, expecially in the xenograft models. This is not a must, but should be further documented in the discussion. Morevoer, olaparib is reported to induce autophagy. The studies done by the authors are not related to this, but this should be mentionned in the discussion as well.
Otherwise, this paper is adequately written, is not the most original, but adds to the body of knowledge on PARP inhibition in HNSCC.
Thank you very much for your comments. Your comments were very inspiring and offered me constructive improvement especially in the relationship between cellular proliferation and autophagy. I will answer to two of your comments.
Regarding comparison of Olaparib alone vs cis-olaparib, I think your opinion that we should compare the effect of Olaparib alone and combination of cisplatin and Olaparib on both EMT/ETM pathway and bone invasion, is quite right. The reason we evaluated these effects by Olaparib alone was just we tried to eliminate prejudice of combinatorial effects from drugs and tried to simply focus on the effect of Parp activity deterioration. Therefore, we added statement about this strategy as follows at L324-327.
In this experiment, we only used Olaparib to evaluate the effects of Parp activity deterioration on EMT/MET pathway and osteogenic invasion, however, combinatorial effects of Olaparib and anti-tumor agents such as cisplatin would be required to proceed to the next step for clinical usage.
Regarding Olaparib and autophagy, the suggestion was quite inspiring to speculate the mechanism of suppression of cellular proliferation in HSC-2 and SAS. We think it would be due to autophagy process caused by Olaparib. Therefore, we hypothesize the mechanism as follows and add the description at L262-271. .
The second most unexpected results were the result of cell proliferation assay, which showed suppressive effect six days after 1 µM Olaparib exposure and SAS and HSC-2 cell lines did not reach confluent (Figure 1-b, c). The mechanism of suppression of cellular proliferation under Olaparib exposure condition could be hypothesized as follows; First, Parp inactivation by Olaparib cause DNA double-strand break which is proved by γ-H2AX expression [41]. Gamma-H2AX is also one of the autophagy markers that regulate cell cycle in cancer cells [41]. Then, accumulation of DNA damage will cause delayed suppressive effects on cellular proliferation, although cell cycle arrest (especially G2/M arrest) was recovered within 48 hours after removing Olaparib, and the cytotoxicity of 1 µM Olaparib was not statistically significant [24].

Reviewer 2 Report
This article is interesting and thoughtful. I think the correction and addition of minor involvement I showed as below will improve your manuscript greater.
Introduction:
L53. 69; “poly(ADP-ribosyl)ation” is your simple word-mistake?
L60, 61, 73, 74, 187, 208, 210, 220, 222, 228, 253, 259, 262, 285, 387, 388; “in vitro”, “in vivo”, and “p” should be written in Italic style.
Results:
L82-85; You should convert these sentences into Materials and Methods section.
Figure 1; In legend, you can delete the abbreviation information of “n.s.; no significance”. Include what the error bar means.
Figure 2 (d-f); You need to clarify what the horizontal axis refers to. Include what the error bar means.
L109; The abbreviation of 3D should be written (three-dimensional) and “Three-dimensional-” in L111 is okay for 3D.
Figure 3; Scale bars are difficult to see. Change the color of bars. Black is better. Include what the error bar means.
Figure 4; “*p<0.05” is not needed in the Figure and legend because all significancy is **p<0.01. Include what the error bar means.
Figure 5; Include what the error bar means.
Figure 7; Ki67 expression index should be calculated and statistically analyzed. The standard method is to select area randomly (3 to 5) and calculate the average positive rates. Data can be plotted or represented by bars.
Discussion:
EMT is just one of the phenomena which drive cancer progression (cell migration). The tumor microenvironment is much heterozygotes. Regarding EMT, there are other concepts for represent tumor diversity: partial EMT and Anaplastic transition (APT). Especially, APT is unique concept. Okuyama et al. reported that there were CK and Vimentin high-expressed epithelial cancer cells in the invasion front of the tumor using multiple immunofluorescent analysis. They found the cells, in which concurrently highly expressed these markers, are in contact with CAFs and they considered that these epithelial cancer cells were affected by CAFs and then got dedifferentiation into more primitive states (anaplasticity). Finally, APT is associated with local recurrence of oral tongue squamous cell carcinoma.
You can include this concept into discussion citing the article below:
Okuyama K, Suzuki K, Yanamoto S, Naruse T, Tsuchihashi H, Yamashita S, Umeda M. Anaplastic transition within the cancer microenvironment in early-stage oral tongue squamous cell carcinoma is associated with local recurrence. Int J Oncol 2018;53(4):1713-1720.
Other, two important things must be explained in discussion section:
Q1; Why can Olaparib affect cell proliferation after Day 6? It took long time to obtain the drug effect. You can add the reason why this delay happened or discuss your opinion.
Q2; In the same experiment, why did HSC-2 cell proliferation in both Olaparib and control group achieve plateau after Day 8? Readers think this is just because cells achieved confluent in Day 8. You should clarify that cells were not confluent in Day 8. For example, you can use 100mm dishes to evaluate cell numbers again.
Materials and Methods:
Q1; How did you obtain these 3 cell lines? Isolated from patients by yourself? If so, the method should be added in this section. Or if not, add where to obtain these each cell lines.
Q2; How did you decide the concentration 1uM of Olaparib? Please add the details of them.
Author Response
This article is interesting and thoughtful. I think the correction and addition of minor involvement I showed as below will improve your manuscript greater.
Thank you very much for your comments. Your comments offered me further-constructively improvement especially in discussion part. I will answer your comments point by point.
- Introduction:
69; “poly(ADP-ribosyl)ation” is your simple word-mistake?
Answer) Both “poly(ADP-ribosyl)ation” were corrected as “poly ADP-ribosylation”.
L60, 61, 73, 74, 187, 208, 210, 220, 222, 228, 253, 259, 262, 285, 387, 388; “in vitro”, “in vivo”, and “p” should be written in Italic style.
Answer) All of “in vitro”, “in vivo”, and “p” were switched to Italic font.
- Results:
L82-85; You should convert these sentences into Materials and Methods section.
Answer) The description was transferred to Material and methods section and stated with the acquisition route of three cell lines at L338-341 as follows;
Three cell lines were used to evaluate the various origins of oral SCC: HSC-2 from oral squamous cell carcinoma in a 69-year-old man, Ca9-22 from gingiva squamous cell carcinoma in a man, and SAS from tongue squamous cell carcinoma. All cell lines were obtained from RIKEN, Tsukuba, Japan, in 2012.
- Figure 1; In legend, you can delete the abbreviation information of “n.s.; no significance”. Include what the error bar means.
Answer) “n.s.; no significance” was deleted and statement for error bar was added.
Figure 2 (d-f); You need to clarify what the horizontal axis refers to. Include what the error bar means.
Answer) Horizontal axis refers to “hour” which was adequately added to Figure2(d-f). The statement for error bar was added.
- L109; The abbreviation of 3D should be written (three-dimensional) and “Three-dimensional-” in L111 is okay for 3D.
Answer) the abbreviation of 3D at L110 was described as “three-dimensional”.
- Figure 3; Scale bars are difficult to see. Change the color of bars. Black is better. Include what the error bar means.
Answer) “200 μM” was deleted for clear visualization, and description was added in the figure legend. The statement for error bar was added.
Figure 4; “*p<0.05” is not needed in the Figure and legend because all significancy is **p<0.01. Include what the error bar means.
Answer) ““*p<0.05” was deleted and the statement for error bar was added.
- Figure 5; Include what the error bar means.
Answer) The statement for error bar was added.
Figure 7; Ki67 expression index should be calculated and statistically analyzed. The standard method is to select area randomly (3 to 5) and calculate the average positive rates. Data can be plotted or represented by bars.
Answer) The mean intensity of Ki67 in each category was analyzed using ImageJ Fiji, and a bar graph was added as Figure 7-b. Accordingly, Figure 7-b and c are converted to Figure 7-c and d, and figure legends regarding new Figure 7-b was also added. Moreover, explanation of Figure 7-b was added and inappropriate sentence was deleted. The protocol of calculation of intensity of Ki67 positivity was also added in material and method section, as follows.
Figure legend: (b) Semi-quantitative analysis of mean intensity of Ki67 positivity in xenograft tumor areas, and significant difference was not observed within each category. Error bars represent standard deviation. *; p < 0.05.
Result section: To quantify the intensity of DAB staining for Ki67 staining in xenograft tumor area, photos of all categories were analyzed by ImageJ Fiji. There was a tendency that the Ki67 intensity of category 0 was less than categories 1-3, however, significant difference was not observed within each category. (L187-191)
Material and Methods section:
4.12 Quantification of the intensity of immunohistochemical positivity
The photographs of the immunohistochemical analysis with Ki67 were analyzed by ImageJ Fiji (National Institutes of Health, Bethesda, MD, USA, available from https://imagej.net/Fiji/Downloads) and the mean intensity of the selected area was calculated. First, all photographs were deconvoluted, and DAB colour version images were chosen from divided into three images. Then, the subjected area was selected and cropped. Subsequently, the mean density of DAB color was calculated and converted from max intensity 255 [63]. (L488-495)
[61] Yamamoto, N.; Kawaguchi, K.; Fujihara, H.; Hasebe, M.; Kishi, Y.; Yasukawa, M.; Kumagai, K.; Hamada,Y. Detection accuracy for epithelial dysplasia using an objective autofluorescence visualization method based on the luminance ratio. Int. J. Oral Sci. 2017, 9, e2-10.
- Discussion:
EMT is just one of the phenomena which drive cancer progression (cell migration). The tumor microenvironment is much heterozygotes. Regarding EMT, there are other concepts for represent tumor diversity: partial EMT and Anaplastic transition (APT). Especially, APT is unique concept. Okuyama et al. reported that there were CK and Vimentin high-expressed epithelial cancer cells in the invasion front of the tumor using multiple immunofluorescent analysis. They found the cells, in which concurrently highly expressed these markers, are in contact with CAFs and they considered that these epithelial cancer cells were affected by CAFs and then got dedifferentiation into more primitive states (anaplasticity). Finally, APT is associated with local recurrence of oral tongue squamous cell carcinoma.
You can include this concept into discussion citing the article below:
Okuyama K, Suzuki K, Yanamoto S, Naruse T, Tsuchihashi H, Yamashita S, Umeda M. Anaplastic transition within the cancer microenvironment in early-stage oral tongue squamous cell carcinoma is associated with local recurrence. Int J Oncol 2018;53(4):1713-1720.
Answer) Thank you very much for this quite attractive suggestion which made us more thoughtful about EMT and MET area. We had not known much about APT, however, Parp inhibition could be hypothesized to be directly involved in APT. Therefore, this hypothesis is also added in the discussion part as follows.
Considering both in vitro and in vivo results, we should notice another concept of anaplastic transition (APT). In the invasion front of carcinoma, there are CK and Vimentin highly expressed epithelial cancer cells, which have contact with Cancer-Associated Fibroblasts (CAFs) and affect each other. Then, these epithelial cancer cells will dedifferentiate into a more primitive status called APT [50]. APT is suggested to have an association with local recurrence of oral tongue squamous cell carcinoma; therefore, Olaparib may have the potency to directly control local recurrence and invasion because of Vimentin attenuation by Olaparib (Figure 5). (L296-303)
- Other, two important things must be explained in discussion section:
Q1; Why can Olaparib affect cell proliferation after Day 6? It took long time to obtain the drug effect. You can add the reason why this delay happened or discuss your opinion.
Q2; In the same experiment, why did HSC-2 cell proliferation in both Olaparib and control group achieve plateau after Day 8? Readers think this is just because cells achieved confluent in Day 8. You should clarify that cells were not confluent in Day 8. For example, you can use 100mm dishes to evaluate cell numbers again.
Answer) Q1 and Q2 would be answered together.
As you estimated, HSC-2 became confluent at day 8, and we had already performed same experience several times. After day 8, Olaparib treated HSC-2 kept approximately 80% confluent and did not appear to proliferate anymore. Moreover, olaparib treated SAS also showed suppressed proliferation and did not reach confluent, too.
We hypothesize the mechanism as follows in the Discussion section at L262-271.
The second most unexpected results were the result of cell proliferation assay, which showed suppressive effect six days after 1 µM Olaparib exposure and SAS and HSC-2 cell lines did not reach confluent (Figure 1-b, c). The mechanism of suppression of cellular proliferation under Olaparib exposure condition could be hypothesized as follows; First, Parp inactivation by Olaparib cause DNA double-strand break which is proved by γ-H2AX expression [41]. Gamma-H2AX is also one of the autophagy markers that regulate cell cycle in cancer cells [41]. Then, accumulation of DNA damage will cause delayed suppressive effects on cellular proliferation, although cell cycle arrest (especially G2/M arrest) was recovered within 48 hours after removing Olaparib, and the cytotoxicity of 1 µM Olaparib was not statistically significant [24].
- Materials and Methods:
Q1; How did you obtain these 3 cell lines? Isolated from patients by yourself? If so, the method should be added in this section. Or if not, add where to obtain these each cell lines.
Answer) All three cell lines were obtained from RIKEN, Tsukuba, Japan, on November 11th, 2012, which was described as follows in Material and Methods section at L338-341.
Three cell lines were used to evaluate the various origins of oral SCC: HSC-2 from oral squamous cell carcinoma in a 69-year-old man, Ca9-22 from gingiva squamous cell carcinoma in a man, and SAS from tongue squamous cell carcinoma. All cell lines were obtained from RIKEN, Tsukuba, Japan, in 2012.
Q2; How did you decide the concentration 1uM of Olaparib? Please add the details of them.
Answer) The concentration of used Olaparib was referred from MTT assay in our previous study, and in the MTT assay, 1uM of Olaparib did not cause significant difference on cell survival. Moreover, several reports showed the effect of Olaparib is dose dependent manner, however, 1uM Olaparib were not significantly effective on another types of cells. Therefore, we thought this concentration would be reasonable to evaluate cellular activity. The process of concentration decision was added as follows at L350-352.
The used olaparib concentration was decided according to previous studies including
the result of MTT assay of our previous study [24][55].
- Weston, V.J.; Oldreive, C.E.; Skowronska, A.; Oscier, D.G.; Pratt, G.; Dyer, M.J.S.; Smith, G.; Powell, J.E.; Rudzki, Z.; Kearns, P.; et al. The PARP inhibitor olaparib induces significant killing of ATM-deficientlymphoid tumor cells in vitro and in vivo. Blood 2010, 116, 4578–4587.
